# Tyk2 Targeting in Immune-Mediated Inflammatory Diseases

**DOI:** 10.3390/ijms24043391

**Published:** 2023-02-08

**Authors:** Lluís Rusiñol, Luis Puig

**Affiliations:** Department of Dermatology IIB Sant Pau, Hospital de la Santa Creu i Sant Pau, Universitat Autònoma de Barcelona, 08041 Barcelona, Spain

**Keywords:** inflammatory diseases, psoriasis, treatment

## Abstract

The Janus kinase (Jak)/signal transducer and activating protein (STAT) pathways mediate the intracellular signaling of cytokines in a wide spectrum of cellular processes. They participate in physiologic and inflammatory cascades and have become a major focus of research, yielding novel therapies for immune-mediated inflammatory diseases (IMID). Genetic linkage has related dysfunction of Tyrosine kinase 2 (Tyk2)—the first member of the Jak family that was described—to protection from psoriasis. Furthermore, Tyk2 dysfunction has been related to IMID prevention, without increasing the risk of serious infections; thus, Tyk2 inhibition has been established as a promising therapeutic target, with multiple Tyk2 inhibitors under development. Most of them are orthosteric inhibitors, impeding adenosine triphosphate (ATP) binding to the JH1 catalytic domain—which is highly conserved across tyrosine kinases—and are not completely selective. Deucravacitinib is an allosteric inhibitor that binds to the pseudokinase JH2 (regulatory) domain of Tyk2; this unique mechanism determines greater selectivity and a reduced risk of adverse events. In September 2022, deucravacitinib became the first Tyk2 inhibitor approved for the treatment of moderate-to-severe psoriasis. A bright future can be expected for Tyk2 inhibitors, with newer drugs and more indications to come.

## 1. Introduction

The intracellular Janus Kinase/signal transducer and activator of transcription (Jak-STAT) pathways play a role in intracellular signaling of cytokines in a wide variety of cellular processes and are important in both normal and pathological states such as immune-mediated inflammatory diseases (IMID), including psoriasis and psoriatic arthritis, among others) [1,2,3].

The Jak family (Jak1, Jak2, Jak3, and Tyk2) comprises receptor-associated tyrosine kinases that act within the cell as signal transducers [4,5]. Activation of the Jak pathway starts with the coupling of a circulating cytokine—e.g., interleukin (IL)-23—to its cell surface receptor, triggering a conformational change of the receptor, which leads to the activation and recruitment of two Jaks. Jak dimers are composed of two different Jak molecules, except Jak2, which can combine with itself. These Jak dimers phosphorylate the receptor, allowing the attachment, phosphorylation, and eventual dimerization of STAT proteins (STAT1, STAT2, STAT3, STAT4, STAT5a, STAT5b, and STAT6) (Figure 1). Activated STAT proteins combine to form dimers and can translocate to the nucleus where they can act as transcription factors, upregulating the genes responsible for production of proinflammatory cytokines and growth factors, or regulate the behavior of other intracellular proteins [1,3,6,7,8].

The Jak/STAT pathway is a therapeutic target in various IMID, since the immune response is coordinated and regulated by soluble mediators corresponding to proinflammatory cytokines in most cases. Jak inhibitors are small molecules that diminish the intracellular transduction of the Jak/STAT pathway, usually by inhibiting the kinase activity of Jak. Due to their small molecular size, Jak inhibitors can be administered orally or topically [1].

Treatment with Jak inhibitors is generally associated with mild to moderate side effects, usually infections of the upper respiratory tract, urinary tract, and gastrointestinal tract, but in some patient populations the risks may overcome the benefits of treatment. Because of the relative non-selectivity of the first generation of Jak inhibitors, most adverse events are considered class specific. Inhibition of Jak1 has been associated with increases in serum levels of triglycerides, total cholesterol, low-density lipoprotein cholesterol, and high-density lipoprotein cholesterol. On the other hand, Jak2 inhibition can interfere with erythropoiesis, myelopoiesis, and platelet activation, leading to anemia, neutropenia, thrombocytopenia, and thromboembolic events. Jak3 expression is restricted to hematopoietic cells and exclusively associated with only the common γc receptor subunit of interleukins regulating lymphocyte activation, function, and proliferation. Tyk2 inhibition can increase the risk of herpesvirus, staphylococcal, and mycobacterial infections [1,9].

## 2. Jak-STAT Signaling and Inhibition

The Jak-STAT pathway mediates downstream signaling of receptors for type I and II cytokines, such as IL-6, IL-10, IL-12, IL-22, IL-23, and interferon (IFN)-α, IFN-β, and IFN-γ (Figure 2). As mentioned before, Jaks associate in pairs. Jak1 pairs with Jak2, Jak3, and Tyk2, leading to the downstream transduction of signals generated by receptors of cytokines such as IFN-α, IFN-γ, IL-2, IL-4, IL-6, IL-7, IL-10, and IL-15 [10,11]. Mice lacking Jak1 have highly impaired lymphopoiesis and inadequate IFN (types I and II) responses, which results in mortality. Jak3 pairs only with Jak1, transducing signals from cytokines sharing the common cytokine receptor γ chain, such as IL-2, IL-4, IL-7, IL-9, IL-15, and IL-21; thus, Jak3 signaling is essential for lymphocyte development [10]. Jak2 forms a homodimer when pairing with itself or a heterodimer with Tyk2. Homodimers of Jak2 mediate signal transduction downstream of IL-3 and hormone-like receptors, such as erythropoietin, growth hormone, prolactin, thrombopoietin, leptin, and granulocyte-macrophage colony-stimulating factor [10], whereas heterodimers of Jak2 and Tyk2 connect with receptors for type I interferons (IFN), IL-12, and IL-23 [12,13].

As shown in Figure 2, cytokine receptors interact with a particular pair of Jaks, which, in turn, interact with some of the existent STAT proteins. Hence, depending on the targeted cytokine, the choice of Jak inhibitor will be different. For example, atopic dermatitis is associated with overexpression of IL-4 and activation of its signaling pathway, which is mediated by the Jak1/Jak3 pair [14]. Conversely, signaling of IL-12 and especially IL-23, which is pathogenetically related to psoriasis, is mediated by Tyk2/Jak2; selective inhibition of Tyk2 avoids interference with multiple Jak2 mediated pathways and potential hematopoietic or thromboembolic adverse events [14,15].

Some Jak inhibitors are more selective than others in their targeting of the active regions in Jak catalytic (JH1) domains. First-generation Jak inhibitors often target two or three separate Jaks and can offer a wide range of therapeutic benefits; however, they may be more likely to cause adverse events than Jak inhibitors from later generations [13,16].

Currently approved therapeutic indications of Jak inhibitors include polycythemia vera (ruxolitinib), myelofibrosis (fedratinib, pacritinib, ruxolitinib), rheumatologic diseases such as rheumatoid arthritis (tofacitinib, baricitinib, peficitinib, filgotinib), ankylosing spondylitis (tofacitinib, upadacitinib), juvenile idiopathic arthritis (tofacitinib) and psoriatic arthritis (tofacitinib, upadacitinib), ulcerative colitis (baricitinib, upadacitinib, tofacitinib), atopic dermatitis (upadacitinib, baricitinib, abrocitinib, and topical delgocitinib in some countries), alopecia areata (baricitinib), and psoriasis (deucravacitinib) [17]. Only the indications of Jak inhibitors related to psoriatic disease will be specifically mentioned below, whereas Tyk2 inhibitors will be discussed extensively in Section 4 of this manuscript.

Tofacitinib, a pan-Jak inhibitor that predominantly targets Jak1 and Jak3, is currently approved for treatment of psoriatic arthritis at the dose of 5 mg twice daily; in phase 3 clinical trials for psoriasis, the 10 mg twice daily regimen was not inferior to etanercept [18,19], but FDA approval was declined based on long term safety issues. Filgotinib, a Jak1 inhibitor [10], and upadacitinib, a Jak1 inhibitor with partial selectivity for Jak2, have also been approved for the treatment of psoriatic arthritis [20].

## 3. Tyk2 Signaling and Pathogenetic Implications

Genetic linkage studies have established a connection between dysfunctional Tyk2 mutations and protection from psoriasis [3,21]. This can be explained by the implication of Tyk2 in multiple pathways related to psoriasis. The inflammatory environment of active psoriatic skin lesions is characterized by the expression of Th1- and Th17-related cytokines, and type I IFN, IL-6, IL-10, IL-12, IL-22, and IL-23 are only a few of the pathogenetically relevant cytokines with signaling pathways affected by Tyk2 loss-of-function mutations (Figure 3) [8,10,22,23]. Inhibiting Tyk2 activation might be associated with an ideal balance between efficacy and safety because individuals with deactivating genetic variants of Tyk2 are highly protected from some IMIDs but do not exhibit an increased risk of hospitalization for mycobacterial, viral, or fungal infections [23,24,25,26,27].

In psoriasis, hidradenitis suppurativa, and other IMIDs, chronic inflammation is initiated and maintained by IL-12 and IL-23 [28,29,30]. IL-12 and IL-23 receptor signal transduction pathways are mediated by the Tyk2/Jak2 heterodimer. IL-12 is necessary for the growth of Th1 cells, which generate pro-inflammatory cytokines such as tumor necrosis factor (TNF)-α and IFN-γ. Th17 cell growth and survival are regulated by IL-23 and keratinocyte growth and activation are enhanced by a combination of cytokines produced by Th1 and Th17 cells [28,31,32].

Tyk2 can also pair with Jak1 to participate in downstream pathway signal transduction of type I IFN receptors, which induce potent antiviral mechanisms [10]. IFN-α and IFN-β are produced rapidly and in great quantities by a variety of cell types, particularly plasmacytoid dendritic cells, during proinflammatory conditions such viral infections [33]. They are also implicated in proinflammatory processes relevant in psoriasis, such as dendritic cell maturation and activation, polarization of Th1 and Th17 cells, impairment of the T-cell regulatory function, and enhancement of the activation of B cells and their antibody production [10,33].

Redundancy in Jak/STAT pathways is common to all Jak combinations except homodimeric Jak2, but the activation of Tyk2 due to IL-12, IL-23, and type I IFN results in STAT-dependent transcription and inflammatory responses different from those occurring subsequent to activation of Jak1, Jak2, and Jak3 [33,34,35,36].

Multiple studies have established that Tyk2 is mandatory for IL-12 and IL-23 signaling. Tyk2-knockout mice do not show epidermal hyperplasia after IL-23 activation, as opposed to Tyk2-positive wild mice; moreover, it has been noted that IL-23 in Tyk2-positive lymphocytes induces dose-dependent secretion of IL-17 and IL-22, which has not been observed in Tyk2-knockout lymphocytes [37].

The need for Tyk2 for type I IFN signaling is less clear, with some contradictory data [38,39]. Majoros et al. reported that Tyk2 was essential for IFN-α signaling, since Tyk2-deficient human cells did not respond to IFN-α [40]. However, in Tyk2-knockout mice, type I IFN signaling was reduced but not suppressed [39]. Therefore, Tyk2 seems to be necessary for type I IFN signaling in human cells, but not in mice cells. 

Additionally, Tyk2 signaling has been linked to the immune system response to the IL-10 family of cytokines. Numerous immune cell types release IL-10, which upon binding to its receptor activates Jak1 and Tyk2, leading to a wide range of immunosuppressive and immunostimulatory effects [38,41,42]. Some of the immunosuppressive effects include inhibition of multiple processes, such as nuclear translocation of the nuclear factor kappa light chain enhancer of activated B cells (NF-kB), IFN-α- and IFN-γ-induced gene transcription, expression of major histocompatibility complex class II molecules by activated dendritic cells and macrophages, and T-cell activation and proliferation [38,41,42,43]. Therefore, clinical studies were performed trying to assess the efficacy of recombinant IL-10 to treat autoimmune diseases. Unfortunately, results were less than encouraging [41]. 

On the other hand, according to several studies, IL-10 promotes humoral immune responses by enabling B cells to differentiate, proliferate, and survive, as well as by stimulating them to produce antibodies [41,42,44]. In systemic lupus erythematosus, high levels of IL-10 expression are considered pathogenic, and its inhibition would be beneficial [41]. Furthermore, IL-10 has been linked to contradictory effects in some cell types, such as natural killer cells, depending on the cellular context [41]. IL-22, an IL-10 family member generated in skin and gastrointestinal tract epithelium, activates Jak1 and Tyk2 and contributes to the epithelial integrity, primarily by enhancing barrier function and triggering the synthesis of antimicrobial peptides. On the other hand, IL-22 also promotes the synthesis by epithelial cells of chemokines that may contribute to tissue damage and gastrointestinal inflammation. This, together with the pathogenetic involvement of IL-12 and IL-13, provides the mechanistic bases for clinical research of Tyk2 inhibitors in the treatment of bowel inflammatory diseases [38,43,44]. 

Type I IFNs can cause monocytes to differentiate into antigen-presenting dendritic cells, the purported key mechanism by which these cells control the activity of autoreactive B and T cells in autoimmune disorders such as lupus and dermatomyositis [34,44]. Additionally, blocking type I IFN receptor (IFNAR) activation with an anti-IFNAR antibody (anifrolumab) lowers disease activity in systemic lupus erythematosus (SLE) patients [34,45]. Whole blood from 31 SLE patients was treated for 5 h with either deucravacitinib or an anti-IFNAR antibody, and the effect on type I IFN-regulated genes was assessed by quantitative PCR. Deucravacitinib decreased the expression of type I IFN-regulated genes that are part of the IFN profile enhanced in SLE patients in this ex vivo test [45]. Deucravacitinib was as effective at the blocking the anti-IFNAR antibody and the response was dose-dependent, with 12 mg twice daily providing almost complete inhibition of CXCL10, ISG20, and IFI27 [45]. These types of I IFN-regulated genes are overexpressed in diseases such as SLE, Sjögren syndrome, and systemic sclerosis [34,44,46].

## 4. Tyk2 Pharmacologic Inhibition

As previously mentioned, Tyk2 is an important mediator in pro-inflammatory signaling, and its inhibition does not seem to carry an increased risk of serious infections [3,33]. Hence, multiple Tyk2 inhibitors are being evaluated for the treatment of inflammatory diseases such as psoriasis, psoriatic arthritis, hidradenitis suppurativa, inflammatory bowel disease, dermatomyositis, and SLE [10,29,30]. Tyk2 inhibitors currently include 21 different molecules, but only 3 of them are selective or predominantly Tyk2 inhibitors; the first includes deucravacitinib (Bristol Myers Squibb, New york, New York, US), ropsacitinib (Priovant Therapeutics, New york, New York, US), BMS-986202 (Bristol Myers Squibb, New york, New York, US), whereas brepocitinib (Priovant Therapeutics, New york, New York, US) and SAR-20347 (ChemScene, Monmouth Junction, NJ, US), albeit potent, are not considered to be selective Tyk2 inhibitors [47] (Table 1). Ghoreschi and colleagues published, in 2021, a Tyk2 revision, focalizing on deucravacitinib [15]. Hereby, we provide a wider revision of Tyk2 inhibitors and their latest results.

### 4.1. Brepocitinib (PF-06700841)

Brepocitinib is a dual Tyk2/Jak1 inhibitor that binds to the active sites in the catalytic domains of Tyk2 and Jak1, with less affinity for Jak2, and even minor selectivity for Jak3 [48,49]. Therefore, adverse events related to Jak2 and Jak3 inhibition are less likely to occur [48]. In a phase 1 trial, brepocitinib was well tolerated by both healthy volunteers and patients with psoriasis, but decreases in platelet and reticulocyte counts occasionally occurred and have been accounted for by some degree of Jak2 inhibition [10,48]. 

#### 4.1.1. Psoriasis

Following a phase 1 clinical trial [50], the efficacy and safety of brepocitinib in psoriasis were evaluated in a phase 2a trial with a four-week induction period of 30 mg once daily, 60 mg once daily, or placebo, followed by an eight-week maintenance phase with brepocitinib 10 mg once day, 30 mg once daily, 100 mg once weekly, or placebo (ClinicalTrials.gov identifier: NCT02969018) [51]. Decreases in Psoriasis Area and Severity Index (PASI) at week 12 (the primary endpoint) were significantly greater in patients who received brepocitinib than in those who received placebo; the greatest change from baseline was observed in the 30 mg once daily continuous treatment group. Treatment was well tolerated and no herpes zoster infections were reported. 

Brepocitinib treatment appeared to lower the expression of several inflammatory genes and cellular pathways associated with the pathogenesis of psoriasis, according to biomarker analyses. In psoriasis skin biopsy specimens, brepocitinib inhibited IL-17A/F and IL-12B expression more rapidly than tofacitinib (2 weeks vs. 4 weeks) [44]. The clinical and molecular improvement with brepocitinib was faster and more complete, with significant reduction in markers of keratinocyte activation, epidermal thickness, KRT16, and Ki-67 expression; furthermore, clinical improvement of psoriasis was associated with a decrease in the immune cell infiltrates CD3^+^/CD8^+^ T cells and CD11c dendritic cells [44,52].

#### 4.1.2. Alopecia Areata

Brepocitinib has also been compared to ritlecitinib—an inhibitor of Jak3 and members of the Tec tyrosine kinase family—and placebo for the treatment of alopecia areata. Severity of Alopecia Tool (SALT)_30_ at 24-weeks of treatment was achieved by 64% of the patients under brepocitinib treatment vs. 50% of patients under ritlecitinib and only 2% of patients on placebo [53]. The improvement in SALT scores was positively associated with expression of Th1 markers and negatively associated with expression of hair keratins [54,55].

#### 4.1.3. Atopic Dermatitis

Finally, topical brepocitinib has been tested for the treatment of atopic dermatitis. In 2019, a phase 2b clinical trial evaluated the efficacy and safety of topical brepocitinib in mild to moderate atopic dermatitis [48,56]. A total of 292 individuals were enrolled and randomized. Brepocitinib 1% once daily and brepocitinib 1% twice daily achieved significant reductions in Eczema and Area Severity Index (EASI) score, compared to the respective vehicle, with no serious adverse events [56].

#### 4.1.4. New Potential Therapeutic Indications

Development of oral brepocitinib has been discontinued for most of its potential indications, including psoriasis, psoriatic arthritis, vitiligo, ulcerative colitis, hidradenitis suppurativa, and Crohn’s disease. In June 2022, Pfizer licensed oral and topical global development rights and US and Japan commercial rights of brepocitinib to Priovant; a phase 2b trial of oral brepocitinib in SLE [57] (trialsearch.who.int/EUCTR2018-004175-12-PL), a phase 2 trial of oral brepocitinib in adults with active non-infectious non-anterior uveitis (NCT05523765), a phase 3 trial of oral brepocitinib in dermatomyositis (NCT05437263), and an investigator-initiated trial of oral brepocitinib in cicatricial alopecia (NCT05076006) are currently ongoing. Phase 2 clinical trials of topical brepocitinib have also been stopped [10,52,57].

### 4.2. Ropsacitinib (PF-06826647)

The oral Tyk2/Jak2 inhibitor ropsacitinib binds to the active site in the catalytic domain (JH1) of each kinase. Despite inhibiting Jak2, ropsacitinib possesses more selectivity for Tyk2 than brepocitinib.

#### 4.2.1. Phase 1 Clinical Trials

Ropsacitinib had an acceptable safety profile in two phase 1 randomized, double-blind, placebo-controlled trials in healthy volunteers and patients with moderate to severe plaque psoriasis [58,59,60]. Ropsacitinib significantly decreased disease activity, as determined by the PASI 75 (≥75% reduction from baseline in Psoriasis Area and Severity Index) response, body surface area, and target plaque severity score after 4 weeks of treatment in the previously mentioned phase 1 study. In addition, this study detected a significant reduction in the IL-17A and IL-17F expression at week 4 of treatment [58]. Increased serum creatinine levels, elevated alanine aminotransferase levels, and headache were the most frequent adverse events in patients with plaque psoriasis. No significant adverse events, fatalities, dose reductions, or temporary discontinuations occurred in individuals receiving ropsacitinib treatment; all side effects were minor [59,60]. 

#### 4.2.2. Phase 2 Clinical Trials

Ropsacitinib phase 2 clinical trials on psoriasis and hidradenitis suppurativa have been completed, but results of the latter (NCT04092452) have not yet been published. The phase 2b clinical trial evaluating ropsacitinib efficacy and safety compared to placebo enrolled and treated up to 178 participants. They were randomized to receive once-daily oral placebo, or ropsacitinib at 50 mg, 100 mg, 200 mg, and 400 mg [61]. The 200 mg and 400 mg groups showed a better proportion of PASI90 (≥90% reduction from baseline in Psoriasis Area and Severity Index) response compared to placebo at week 16 [61]. Ropsacitinib was well tolerated at week 40 of follow-up and only 18 patients discontinued the treatment due to treatment-related adverse events [61]. 

The phase 2b clinical trial of ropsacitinib in patients with ulcerative colitis was withdrawn (NCT04209556) [57]. Currently, there is no ongoing clinical trial of ropsacitinib, which has also been licensed by Pfizer to Priovant [62].

### 4.3. Deucravacitinib

Deucravacitinib is an oral Tyk2 inhibitor that binds to the regulatory (JH2 pseudokinase) domain rather than the active (ATP-binding) site in the catalytic (JH1) domain of Tyk2, where other Tyk2/Jak1-3 inhibitors attach (Figure 4) [3,34,63,64]. The allosteric binding of deucravacitinib locks the regulatory (JH2) domain into an inhibitory contact with the catalytic domain. This leads to the inactivation of Tyk2, inhibiting receptor-mediated activation and subsequent signal transduction [34].

Deucravacitinib is highly selective for Tyk2, with little to no activity against Jak1–3. In cell-based assays, deucravacitinib has more than 100-fold greater selectivity for Tyk2 over Jak1/3 and more than 2000-fold greater selectivity for Tyk2 over Jak2 (half maximal inhibitory concentration [IC50] data) [34,64,65].

#### 4.3.1. Psoriasis

Deucravacitinib was evaluated in a randomized, placebo-controlled, dose-ranging 12 week phase 2 trial including 267 adult patients (NCT02931838); the PASI75 response rates at week 12 (the primary endpoint) for deucravacitinib were 68.9% (3 mg twice daily), 66.7% (6 mg twice daily), and 75% (12 mg once daily) vs. placebo (6.7%) [66]. 

Skin biopsy specimens taken after deucravacitinib treatment in this phase 2 trial demonstrated normalization of inflammatory gene expression and inhibition of biologic markers of disease activity, such as the expression of the IL-23/Th17 and type I IFN signaling pathways, which are involved in keratinocyte activation and inflammation in psoriasis [67].

The most common adverse events were mild and included nasopharyngitis, headaches, diarrhea, nausea, and upper respiratory tract infections. No herpes zoster infections or cardiovascular events, which are adverse events of special interest for Jak inhibitors, were reported. One patient was diagnosed of melanoma during treatment [66]. Patients receiving deucravacitinib had no significant alterations of hematologic parameters (lymphocyte, natural killer cell, neutrophil, and platelet counts and hemoglobin levels), serum lipids (high-density lipoprotein and low-density lipoprotein cholesterol), creatinine, immunoglobulins, or liver enzymes [10,66,68]. Regarding dyslipidemia, its development as an adverse event is partially related to the impact on IL-6 signaling, which is regulated by Jak; since deucravacitinib does not interact with Jak1 in human cells, this adverse event should not be expected and the same applies to hematologic alterations occurring when Jak2 is inhibited. Therefore, deucravacitinib reduces the potential risk of abnormalities in laboratory parameters [12,16,34,66].

A post-hoc analysis of the same phase 2 clinical trial assessed additional clinical and quality of life (QoL) outcomes of the three most efficacious regimens (3 mg twice daily, 6 mg twice daily and 12 mg once daily) and placebo [69]. Early improvement was noted at week 4 in patients under deucravacitinib, and clinical responses, including PASI and Body surface Area (BSA), showed similar trends to QoL measured by Dermatology Life Quality Index (DLQI) [69]. In patients combined from the three deucravacitinib regimens, DLQI 0/1 response was achieved by 17.9% at week 4, 41.8% at week 8, and 55.2% at week 12. DLQI 0/1 response rates on the placebo group were 6.7%, 4.4%, and 4.4%, respectively [69]. 

These results indicated significant commercial potential for this new oral treatment for psoriasis, where apremilast has a leading market position in the USA and other countries; apparently apremilast had to be sold to expedite the Federal Trade Commission approval of the merger between Bristol Myers Squibb and Celgene [70]. 

Results from two phase 3 clinical trials comparing deucravacitinib to placebo and apremilast have been published. POETyk PSO-1 was a 52 week phase 3 trial in which participants were randomized 2:1:1 to receive deucravacitinib 6 mg every day (*n* = 332), placebo (*n* = 166), or apremilast 30 mg twice a day (*n* = 168) (NCT03624127) [71]. At week 16, response rates were significantly higher with deucravacitinib versus placebo or apremilast for PASI75 194 [58.4%] vs. 21 [12.7%] vs. 59 [35.1%] and static Physician’s Global Assessment score of 0 or 1 (sPGA 0/1): 178 [53.6%] vs. 12 [7.2%] vs. 54 [32.1%]. Efficacy improved beyond week 16 and was maintained through week 52. The most common adverse events related to deucravacitinib were nasopharyngitis and upper respiratory tract infection [71]. 

In POETyk PSO-2 (NCT03611751), a total of 1020 patients were randomized: 511 received deucravacitinib 6 mg once daily, 255 placebo, and 254 patients were treated with apremilast 30 mg twice daily [72]. Patients on deucravacitinib treatment showed better results than placebo and apremilast. At week 16, PASI75 was achieved by 53% of the patients treated with deucravacitinib, compared to 9.4% and 39.8% of the patients treated with placebo and apremilast, respectively. At week 52, efficacy with continuous deucravacitinib was maintained. The most frequent adverse event was nasopharyngitis, no meaningful laboratory changes were detected, and discontinuations due to adverse events were infrequent [72].

The efficacy and safety of deucravacitinib in treating moderate to severe psoriasis, scalp psoriasis, nail psoriasis, and in pediatric patients with moderate to severe psoriasis are being examined in phase 3 and 4 clinical trials (NCT04036435, NCT05478499, NCT05124080, and NCT04772079, respectively). Moreover, a phase 4 observational post-marketing surveillance of adverse events in patients with psoriasis in Japan is expected to start recruiting patients in the near future (NCT05633264). Finally, an adherence clinical trial in patients with psoriasis is expected to start recruiting soon (NCT05570955).

Deucravacitinib was first approved in the USA on 9 September 2022 for the treatment of moderate to severe plaque psoriasis in adults who are candidates for systemic therapy or phototherapy. Subsequently, on 26 September 2022, it was approved in Japan for the treatment of plaque psoriasis, generalized pustular psoriasis, and erythrodermic psoriasis. Currently, deucravacitinib is under consideration by the European Medicines Agency [73].

#### 4.3.2. Psoriatic Arthritis

A phase 2 clinical trial (NCT03881059) evaluated the efficacy and safety of deucravacitinib compared to placebo in patients with active psoriatic arthritis [74]. A total of 203 participants were randomized in three groups: placebo, deucravacitinib 6 mg once daily, and deucravacitinib 12 mg once a day. The primary endpoint was American College of Rheumatology-20 (ACR-20) response at week 16. The ACR-20 response was significantly higher in the 6 mg (52.9%) and 12 mg (62.7%) groups compared to placebo. No serious adverse events, herpes zoster, or cardiovascular events were reported. Laboratory measures were not different to placebo group. The most common adverse events reported in deucravacitinib-treated patients were nasopharyngitis, upper respiratory tract infection, sinusitis, bronchitis, rash, headache, and diarrhea [74]. Currently, two phase 3 clinical trials for active PsA are recruiting patients (NCT04908202 and NCT04908189). 

#### 4.3.3. Other Therapeutic Indications

Deucravacitinib is being evaluated in a number of phase 2 and 3 trials for alopecia areata (NCT05556265), moderate to severe ulcerative colitis (NCT03934216), Crohn’s disease (NCT04877990), active SLE (NCT05617677), active discoid, and subacute cutaneous lupus erythematosus (NCT04857034). Results of these clinical trials are not available yet, except for those of the phase 2 clinical trial in ulcerative colitis, where deucravacitinib failed to meet its primary or secondary efficacy endpoints at week 12 [75].

### 4.4. BMS-986202

Structural and molecular variations applied on deucravacitinib led to multiple possible new drugs, and pharmacokinetic studies eventually led to further development of BMS-986202 as a potential new oral Tyk2 inhibitor binding to the regulatory JH2 pseudokinase domain [76]. To evaluate the pharmacodynamic responses of BMS-986202, its ability to inhibit IFN-γ production induced by the stimulation with IL-12/IL-18 was tested in mice. A dose-dependent reduction in IFN-γ production was detected, with an up to 80% reduction of IFN-γ levels with the highest dose [76].

Subsequently, the potential efficacy of BMS-986202 was evaluated on psoriasis, ulcerative colitis, and SLE. A well-established psoriasis-like model in mice is the induction of skin acanthosis with IL-23 injections. In this model, BMS-986202 inhibited acanthosis in a dose–response manner and the 30 mg/kg once-daily dose showed at least equivalent efficacy to ustekinumab, which was used as a positive control in the study [76]. The murine model of ulcerative colitis consists of a single injection of an anti-CD40 antibody in severe combined immune-deficient (SCID) mice, resulting in induced colitis. BMS-986202 was orally dosed in these mice for 6 days, and histopathologic studies showed a dose-dependent inhibition of colitis, with 25 and 60 mg/kg doses yielding equivalent efficacy to the anti-p40 antibody that was used as a positive control [76]. In a spontaneous lupus model (NZB/W mice), BMS-986202 inhibited development of anti-dsDNA titers and severe proteinuria in a dose-dependent manner [76].

As regards human studies, a phase 1 clinical trial evaluating BMS-986202 safety, tolerability, pharmacokinetics, and pharmacodynamics has been completed, but results have not been posted yet (NCT02763969).

### 4.5. SAR-20347

SAR-20347 is an oral inhibitor of Tyk2 and Jak1 with selectivity over Jak2 and Jak3. Preclinical assays demonstrated that SAR-20347 inhibited IL-12, IL-23 and IFN-α signaling [22]. Both Tyk2 mutant mice and mice treated with SAR-20347 showed significant reduction of IL-6 and IL-17 in imiquimod-induced skin lesions, but only SAR-20347-treated mice presented reduced levels of IL-23, decreased keratinocyte proliferation and improved clinical score [22]. In addition, SAR-20347-treated mice manifested lower IL-17 gene expression compared to Tyk2 mutant mice [22]. In this model, Works et al. demonstrated that SAR-20347 treated mice showed an almost complete loss of IL-22 gene expression in skin lesions, and they postulate that SAR-20347 would impair the ability of Th17 and γδ cells to induce IL-22 [22]. IL-22 is required for development of autoreactive Th17 cells [77,78]. However, not only IL-22 production was impaired, since IL-22 signaling was also affected in vitro in a human colonic cell line, and STAT3 phosphorylation dependent of IL-22 was completely blocked [22]. 

Blocking both Tyk2 and Jak1 in this study was more effective than inhibition of Tyk2 alone at reducing psoriasis-like disease severity, keratinocyte proliferation, as well as IL-23, IL-17, IL-6, IL-22, and antimicrobial peptide gene expression, and the authors postulate that targeting a combination of Jak1 and Tyk2 using an orally available inhibitor may be a viable approach for treating psoriasis, but currently there are no ongoing or completed clinical trials for SAR-20347 [22].

## 5. Discussion

The Jak/STAT pathway is a signaling keystone in multiple physiological processes, but also in pathological conditions, such as cancer or inflammatory diseases including psoriasis, rheumatoid arthritis, SLE, and other IMIDs [1,4]. Consequently, pharmacologic intervention on this pathway is an area of intensive research [33,79]. 

As regards inflammatory dermatoses, some Jak inhibitors (e.g., baricitinib, upadacitinib, and abrocitinib) have recently gained approval for the treatment of atopic dermatitis [14], and multiple new drugs are being developed with encouraging results. Tyk2 participates in the signaling of multiple regulatory and effector cytokines involved in psoriasis [10,44], and Tyk2 dysfunctional mutations are protective for psoriasis [10,22]. Thus, Tyk2 is an attractive druggable target, and multiple Tyk2 inhibitors are being investigated [10,13,22]. Deucravacitinib has become the first Tyk2 inhibitor approved for moderate to severe psoriasis treatment [73], but potential indications in other IMIDs are multiple and the therapeutic potential of Tyk2 inhibitors are not limited to psoriatic disease. 

Most Jak and Tyk inhibitors are orthosteric, binding at the active tyrosine kinase site and competing with ATP, as opposed to the allosteric inhibition of deucravacitinib and BMS-986202, which bind to the pseudokinase JH2 (regulatory) domain of Tyk2, providing unparalleled selectivity [10,65,73]. As demonstrated in clinical trials of deucravacitinib, this unique mechanism of action limits the side effects of deucravacitinib and, expectedly, BMS-986202 [10,22,71,72]. 

Deucravacitinib phase 3 clinical trials (POETyk PSO-2 and POETyk PSO-1) demonstrated efficacy in psoriasis, with almost 70% of the patients achieving PASI75 response by week 24 [10,69,71,72]. This efficacy level falls behind the most efficacious currently available biologic agents, but when compared to apremilast, the PASI75 response rates of deucravacitinib at week 16 were 23% higher. Studies comparing deucravacitinib to other Jak inhibitors and biologics are still lacking, but the efficacy results are remarkable for an oral agent and the safety profile is reassuring. 

More Tyk2 inhibitors may be on their way, such as brepocitinib, ropsacitinib, BMS-986202, and SAR-20347 [10,76]. Safety concerns regarding non-selective Jak/Tyk inhibitors—mostly based on safety issues with tofacitinib and baricitinib in selected patient populations—may be a drawback for future development of some of these drugs in some indications. Brepocitinib and ropsacitinib have been partially disowned by Pfizer, but brepocitinib is still being tested for SLE, dermatomyositis, and non-anterior non-infectious uveitis. There are no ongoing clinical trials for the remaining Tyk2 inhibitors at the time of writing.

## 6. Conclusions

Multiple Jak inhibitors have been approved or are being developed for myeloproliferative disorders and several IMIDs; they represent a valuable therapeutic alternative in atopic dermatitis and deucravacitinib can be considered a first-in-class oral treatment for psoriasis. 

Tyk2 inhibition targets the signaling of several cytokines that are especially relevant in the pathogenesis of psoriatic disease, while not interfering with other physiological responses mediated by other Jaks. Selective allosteric inhibition of Tyk2 seems to avoid some adverse events (dyslipidemia, laboratory abnormalities, thromboembolic events, cancer, or severe infections) that have been observed in some populations of patients treated with non-selective Jak inhibitors, mostly for rheumatological indications. On the other hand, blockade of type I IFN or IL-12 signaling does not appear to be associated with a marked increase in predisposition to severe viral, bacterial, or fungal infections. 

Tyk2-mediated signal transduction lies at the crossroads of multiple pathogenic cytokines, and its specific inhibition may provide effective treatment with comorbid IMIDs, such as psoriasis, psoriatic arthritis, spondyloarthropathies, and inflammatory bowel disease. Furthermore, multiple diseases characterized by overexpression of type I IFN—such as SLE, Sjögren syndrome, or systemic sclerosis—might benefit from selective Tyk2 inhibition. Jak inhibitors mediate in the cornerstone of the inflammatory pathways; hence a single agent can be used to treat different IMIDs in the same patient.

## 7. Future Directions

There are great expectations related to the approval and use of Tyk2 inhibitors. On 9 September 2022, deucravacitinib became the first oral Tyk2 inhibitor approved by the USA Food and Drug Administration (FDA) for the treatment of moderate to severe plaque psoriasis. Two weeks later, the Japanese regulatory agency approved deucravacitinib for the treatment of plaque psoriasis, generalized pustular psoriasis, and erythrodermic psoriasis [73]. Approval by the European Medicines Agency is still pending. With the accrual of real-world experience, the true potential and the safety profile of deucravacitinib will be fully assessed, and the role of this oral treatment in the crowded therapeutic arena of psoriasis will be established. The eventual approval of deucravacitinib in psoriatic arthritis might provide an alternative to currently available Jak inhibitors with perhaps an improved safety profile. Furthermore, deucravacitinib might provide a much-needed therapeutic alternative in other potential dermatologic indications, such as alopecia areata, active SLE, and cutaneous lupus erythematosus. 

Nonetheless, only deucravacitinib has completed phase 3 clinical trials. Despite its efficacy for the treatment of moderate to severe psoriasis, it does not achieve PASI 90 or PASI 100 response rates comparable to those of IL-17 or IL-23 inhibitors, and it has not been directly compared with adalimumab or ustekinumab. Moreover, the safety profile of Jak inhibitors is being currently scrutinized, at least in selected populations, but the safety profile of deucravacitinib is reassuring, including the absence of significant laboratory abnormalities. 

Future development of non-selective Tyk2 inhibitors will be focused on diseases where therapeutic benefit outweighs potential safety concerns, as illustrated by the current indications pursued for brepocitinib (SLE, dermatomyositis, and uveitis). On the other hand, the future seems bright for selective Tyk2 inhibitors that maintain the safety profile of deucravacitinib and provide improved efficacy in psoriasis—perhaps comparable to that of biologic agents targeting the regulatory cytokine IL-23—and in other potential indications.

## Figures and Tables

**Figure 1 ijms-24-03391-f001:**
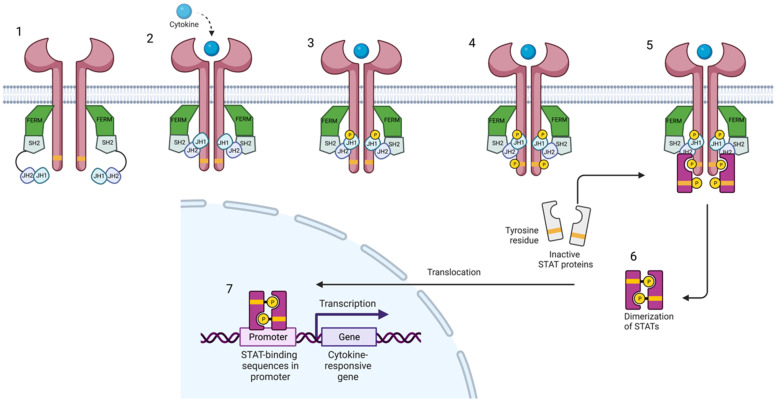
Janus kinase-signal transducer and activator of transcription pathway. Cytokines bind to their receptor, leading to the activation of Jak and their phosphorylation. Subsequently, STAT is phosphorylated and dimerized. Activated STAT dimers translocate to the nucleus and regulate gene transcription and expression. Created with BioRender.com (accessed on 14 January 2023). Abbreviations: FERM, 4.1 ezrin, radixin moesin domain; SH2, Src-homology 2 domain; JH1, jak homology domain 1 (kinase domain); JH2, jak homology domain 2 (pseudokinase domain).

**Figure 2 ijms-24-03391-f002:**
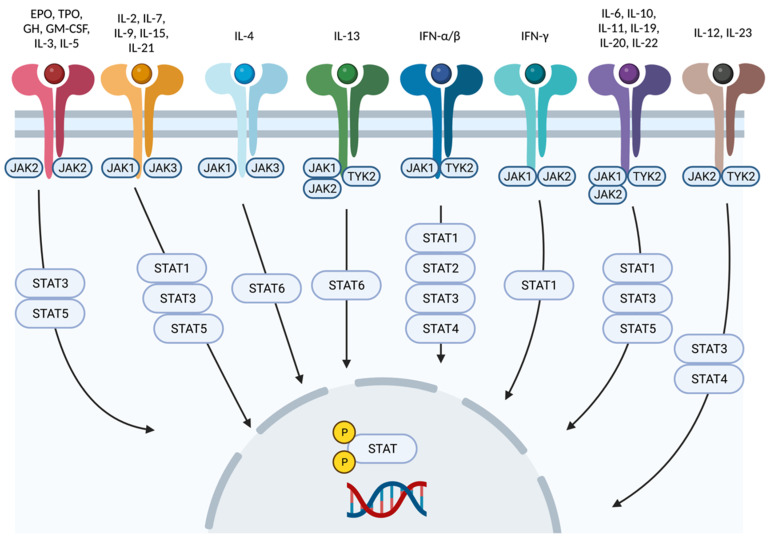
Cytokine signaling mediated by Jak-STAT. Each cytokine interacts with a particular combination of Jak-STAT molecules. Created with BioRender.com (accessed on 14 January 2023). Abbreviations: Jak, Janus kinase; STAT, signal transducer and activator of transcription; Tyk2, tyrosine kinase 2; IFN, interferon; IL, interleukin; EPO, erythropoietin; GH, growth hormone; TPO, thrombopoietin; GM-CSF, granulocyte macrophage colony-stimulating factor.

**Figure 3 ijms-24-03391-f003:**
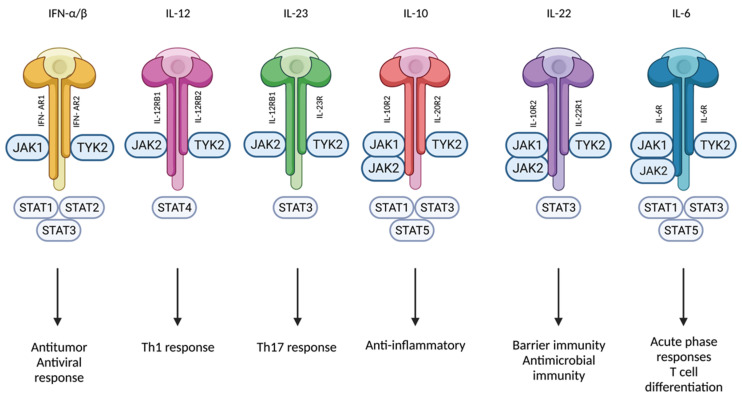
Tyk2 mediates signaling of IFN-α/β, IL-12, IL-23, IL-10, IL-22, and IL-6. Created with BioRender.com (accessed on 14 January 2023). Abbreviations: Tyk2, tyrosine kinase 2; IFN, interferon; IL, interleukin; Jak, Janus kinase; STAT, signal transducer and activator of transcription; Th, T-helper; R, Receptor.

**Figure 4 ijms-24-03391-f004:**
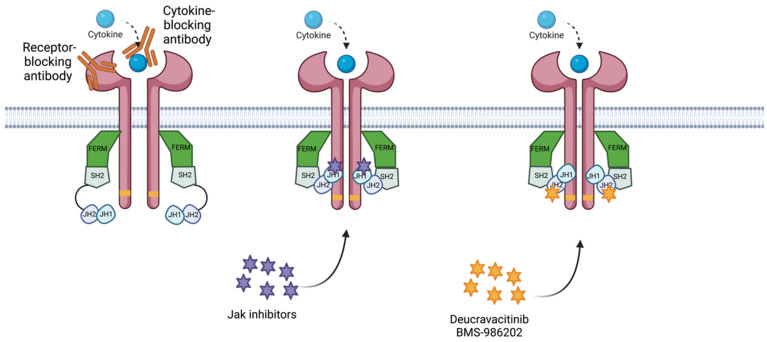
Diagram of mechanism of action for the treatments for IMIDs. The receptor-blocking antibody binds to the receptor, preventing its capacity to bind to the cytokine. The. cytokine-blocking antibody binds to the cytokine, blocking its capacity to bind to the receptor. Jak inhibitors bind to the JH1 domain—kinase domain—impeding adenosine triphosphate (ATP) binding to the JH1 catalytic domain. Deucravacitinib and BMS-986202 bind to the JH2 domain, locking the regulatory (JH2) domain into an inhibitory contact with the catalytic domain (JH1). Created with BioRender.com (accessed on 14 January 2023). Abbreviations: FERM, 4.1 ezrin, radixin moesin domain; SH2, Src-homology 2 domain; JH1, jak homology domain 1 (kinase domain); JH2, jak homology domain 2 (pseudokinase domain).

**Table 1 ijms-24-03391-t001:** Summary of the main characteristics of the Tyk2 inhibitors discussed in this review.

Treatment	Mechanism of Action	Oral and/or Topical	Approved Indications	Tested Indications	Most Frequent Adverse Events	Current Clinical Trials
Brepocitinib	Tyk2/Jak1 inhibitor (binds to the catalytic domain, JH1)	Oral and topical	Not yet	Psoriasis Atopic dermatitis Alopecia areata Discontinued on trials for psoriasis, psoriatic arthritis, vitiligo, ulcerative colitis, hidradenitis suppurativa, and Crohn disease	Nasopharyngitis, upper respiratory tract infection, and headache	Systemic lupus erythematosus Active non-infectious non-anterior uveitis Dermatomyositis Cicatricial alopecia
Ropsacitinib	Tyk2/Jak2 inhibitor (binds to the catalytic domain, JH1)	Oral	Not yet	Psoriasis Hidradenitis suppurativa (results pending) Discontinued on trial for ulcerative colitis	Increased serum creatinine levels, elevated alanine aminotransferase levels, and headache	No ongoing clinical trials
Deucravacitinib	Tyk2 inhibitor (binds to the regulatory domain, JH2)	Oral	Moderate to severe plaque psoriasis (USA, Japan) Generalized pustular psoriasis (Japan) Erythrodermic psoriasis (Japan)	Psoriasis Psoriatic arthritis	Nasopharyngitis, and upper respiratory tract infection * No dyslipidemia	Alopecia areata Ulcerative colitis Crohn’s disease SLE Discoid and subacute cutaneous lupus erythematosus
BMS-986202	Tyk2 inhibitor (binds to the regulatory domain, JH2)	Oral	Not yet	No	Not available	No ongoing clinical trials
SAR-20347	Tyk2/Jak1 inhibitor (binds to the catalytic domain, JH1)	Oral	Not Yet	No	Not available	No ongoing clinical trials

* No dyslipidemia: Deucravacitinib demonstrated no dyslipidemia, conversely to the remaining Tyk2 inhibitors tested.

## Data Availability

No new data were created or analyzed in this study. Data sharing is not applicable to this article.

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
