# Peer review of "Tyk2 Targeting in Immune-Mediated Inflammatory Diseases"

_ijms, 2023, doi:10.3390/ijms24043391_

Round 1

Reviewer 1 Report

The authors reviewed and summarized the current state of scientific knowledge and understanding of the role of the Tyk2 signaling pathway in immune-mediated inflammatory disease. The manuscript was prepared relatively well, and the descriptions were partly sufficient to support the conclusion. However, there are several issues to consider for publication, as follows:

1.      Because this manuscript is quite similar to the review article by Ghoreschi et al. (DOI: 10.1111/ddg.14585), the author must distinguish it.

2.      The title is so broad. If the author wants to focus on skin inflammatory diseases (psoriasis), author need to include and explain the differences regarding the Tyk2 targeting pathway in other skin inflammatory diseases in the introduction part.

3.      I suggest that the author needs to explain the clinical use limitations of Tyk2 inhibitors in the "future directions" part by giving clinical trial data examples.

Author Response

The authors reviewed and summarized the current state of scientific knowledge and understanding of the role of the Tyk2 signaling pathway in immune-mediated inflammatory disease. The manuscript was prepared relatively well, and the descriptions were partly sufficient to support the conclusion. However, there are several issues to consider for publication, as follows:

  1. Because this manuscript is quite similar to the review article by Ghoreschi et al. (DOI: 10.1111/ddg.14585), the author must distinguish it.

As pointed out, both articles address Tyk2 and its inhibition, but we believe that our manuscript entails a wider revision of Tyk2 inhibitors, since Ghoreschi et al, only refer to deucravacinib treatment. Moreover, as their manuscript was published one year ago, deucravacitinib phase 3 clinical trials are not included. These clinical trials demonstrated deucravacitinib superiority to apremilast, being a remarkable result.

We added at line 205:

Ghoreschi and colleagues published in 2021 a Tyk2 revision, focalizing on deucravacitinib. Hereby, we provide a wider revision of Tyk2 inhibitors and their latest results.

  1. The title is so broad. If the author wants to focus on skin inflammatory diseases (psoriasis), author need to include and explain the differences regarding the Tyk2 targeting pathway in other skin inflammatory diseases in the introduction part.

We added this paragraph at the section: 2.Jak signaling and inhibition

As shown in Figure 2, cytokine receptors interact with a particular pair of Jaks, which in turn interact with some of the existent STAT proteins. Hence, depending on the targeted cytokine, the choice of Jak inhibitor will be different. For example, atopic dermatitis is associated with overexpression of IL-4 and activation of its signaling pathway, which is mediated by the Jak1/Jak3 pair. Conversely, signaling of IL-12 and especially IL-23, which is pathogenetically related to psoriasis, is mediated by Tyk2/Jak2; selective inhibition of Tyk2 avoids interference with multiple Jak2 mediated pathways and potential hematopoietic or thromboembolic adverse events.

  1. I suggest that the author needs to explain the clinical use limitations of Tyk2 inhibitors in the "future directions" part by giving clinical trial data examples.

We added this paragraph in section: 7. Future directions

Nonetheless, only deucravacitinib has completed phase 3 clinical trials. Despite its efficacy for the treatment of moderate-to-severe psoriasis, it does not achieve PASI 90 or PASI 100 response rates comparable to those of IL-17 or IL-23 inhibitors, and it has not been directly compared with adalimumab or ustekinumab. Moreover, the safety profile of Jak inhibitors is being currently scrutinized, at least in selected populations, but the safety profile of deucravacitinib is reassuring, including the absence of significant laboratory abnormalities.

Reviewer 2 Report

This review deals with a timely topic and shows good indications for TYK2 inhibitors for the treatment of psoriasis.

Although some parts are very convincing , the structure of the review needs some changes.

E.g. the discussion is mostly a repetition of the clinical data for Deucravacitinib.

Also the parts 4.5. SAR-20347 and 4.4. BMS-986202 after the Deucravacitinib part does not really fut. These go back to mostly mouse data whereas for Deucravacitinib mostly clinical data are presented in that section.

The section about IL-10 seems confusing (starting with: “Additionally, Tyk2 signaling has been linked to the immune system response to the 152 IL-10 family of cytokines.”

It is not entirely clear whether IL-10 inhibition is of benefit or rather a possible side effect? This part need more clarification.

Author Response

This review deals with a timely topic and shows good indications for TYK2 inhibitors for the treatment of psoriasis.

Although some parts are very convincing , the structure of the review needs some changes.

  1. g. the discussion is mostly a repetition of the clinical data for Deucravacitinib.

Paragraphs 3 and 4 of section: 5. Discussion (lines 446-453) have been modified:

As demonstrated in clinical trials of deucravacitinib, this unique mechanism of action limits the side effects of deucravacitinib and, expectedly, BMS-986202 [10,20,70,71].

Deucravacitinib phase 3 clinical trials (POETyk PSO-2 and POETyk PSO-1), demonstrated efficacy in psoriasis, with almost 70% of the patients achieving PASI75 response by week 24 [10,68,70,71]. This efficacy level falls behind the most efficacious currently available biologic agents, but when compared to apremilast, the PASI75 response rates of deucravacitinib at week 16 were 23% higher.

  1. Also the parts 4.5. SAR-20347 and 4.4. BMS-986202 after the Deucravacitinib part does not really fit. These go back to mostly mouse data whereas for Deucravacitinib mostly clinical data are presented in that section.

Since we considered the importance of noting their existence and future availability, we described current evidence for SAR-20347 and BMS-986202.Unfortunately, no clinical trials are available for SAR-20347, and phase 1 clinical trial results for BMS-986202 have not been published yet. Therefore, we presented their preclinical data.

  1. The section about IL-10 seems confusing (starting with: “Additionally, Tyk2 signaling has been linked to the immune system response to the 152 IL-10 family of cytokines.” It is not entirely clear whether IL-10 inhibition is of benefit or rather a possible side effect? This part need more clarification.

We added in section: 3. Tyk2 signaling and pathogenetic implications some further clarification:

Therefore, clinical studies were performed trying to assess the efficacy of recombinant IL-10 to treat autoimmune diseases. Unfortunately, results were less than encouraging [39].

On the other hand, according to several studies, IL-10 promotes humoral immune responses by enabling B cells to differentiate, proliferate, and survive as well as by stimulating them to produce antibodies [39,40,42]. In systemic lupus erythematosus high levels of IL-10 expression are considered pathogenic, and its inhibition would be beneficial [39].

Reviewer 3 Report

The authors present a review about Tyk2 targeting in immune mediated inflammatory diseases. The information they provide is complete and may be of interest to researchers and physicians working in the area of autoimmune diseases.
There is a lot of information about some of the clinical trials in which TyK2 pathway blockers are being used and have shown fewer adverse effects.

It would be advisable for the authors to add a table where the advantages, disadvantages, etc., of the different drugs that they mention during the development of the article are presented in a summarized way, for a faster reference and visualization.

English language and style are fine/minor spell check required.

Finally, I consider that the proposal is an interesting and updated review of what is being tested in the field of treatment of some autoimmune diseases.

Author Response

The authors present a review about Tyk2 targeting in immune mediated inflammatory diseases. The information they provide is complete and may be of interest to researchers and physicians working in the area of autoimmune diseases. There is a lot of information about some of the clinical trials in which TyK2 pathway blockers are being used and have shown fewer adverse effects.

It would be advisable for the authors to add a table where the advantages, disadvantages, etc., of the different drugs that they mention during the development of the article are presented in a summarized way, for a faster reference and visualization.

English language and style are fine/minor spell check required.

Finally, I consider that the proposal is an interesting and updated review of what is being tested in the field of treatment of some autoimmune diseases

Thank you very much for your kind review.

A Table (1) has been added:

Treatment

Mechanism of action

Oral and/or topical

Approved Indications

Tested indications

Most frequent adverse events

Current clinical trials

Brepocitinib

Tyk2/Jak1 inhibitor (binds to the catalytic domain, JH1)

Oral and topical

Not yet

·       Psoriasis

·       Atopic dermatitis

·       Alopecia areata

o    Discontinued on trials for psoriasis, psoriatic arthritis, vitiligo, ulcerative colitis, hidradenitis suppurativa, and Crohn disease

Nasopharyngitis, upper respiratory tract infection, and headache

·       SLE

·       Active non-infectious non-anterior uveitis

·       Dermatomyositis

·       Cicatricial alopecia

Ropsacitinib

Tyk2/Jak2 inhibitor (binds to the catalytic domain, JH1)

Oral

Not yet

·       Psoriasis

·       Hidradenitis suppurativa (results pending)

o    Discontinued on trial for ulcerative colitis

Increased serum creatinine levels, elevated alanine aminotransferase levels, and headache

·       No ongoing clinical trials

Deucravacitinib

Tyk2 inhibitor (binds to the regulatory domain, JH2)

Oral

·       Moderate-to-severe plaque psoriasis (USA, Japan)

·       Generalized pustular psoriasis (Japan)

·       Erythrodermic psoriasis (Japan)

·       Psoriasis

·       Psoriatic arthritis

Nasopharyngitis, and upper respiratory tract infection

*No dyslipidemia

·       Alopecia areata

·       Ulcerative colitis

·       Crohn disease

·       SLE

·       Discoid and subacute cutaneous lupus erythematosus

BMS-986202

Tyk2 inhibitor (binds to the regulatory domain, JH2)

Oral

Not yet

No

No available

·       No ongoing clinical trials

SAR-20347

Tyk2/Jak1 inhibitor (binds to the catalytic domain, JH1)

Oral

Not Yet

No

No available

·       No ongoing clinical trials
